# Sexual Dimorphism in Sex Hormone Metabolism in Human Skeletal Muscle Cells in Response to Different Testosterone Exposure

**DOI:** 10.3390/biology13100796

**Published:** 2024-10-05

**Authors:** Paolo Sgrò, Cristina Antinozzi, Christopher W. Wasson, Francesco Del Galdo, Ivan Dimauro, Luigi Di Luigi

**Affiliations:** 1Endocrinology Unit, Department of Movement, Human and Health Sciences, University of Rome Foro Italico, 00135 Rome, Italy; paolo.sgro@uniroma4.it (P.S.); luigi.diluigi@uniroma4.it (L.D.L.); 2Leeds Institute of Rheumatic and Musculoskeletal Medicine, Faculty of Medicine and Health, University of Leeds, Leeds LS7 4SA, UK; chriswasson85@yahoo.co.uk (C.W.W.); f.delgaldo@leeds.ac.uk (F.D.G.); 3Unit of Biology and Genetics of Movement, Department of Movement, Human and Health Sciences, University of Rome Foro Italico, 00135 Rome, Italy; ivan.dimauro@uniroma4.it

**Keywords:** gender dimorphism, muscle steroidogenesis, myokines, sex-chromosome, sex-related health status

## Abstract

**Simple Summary:**

Skeletal muscle is an endocrine organ capable of responding to and releasing hormones and hormonally active molecules. In particular, sex hormones (e.g., testosterone) specifically regulate metabolic activity, growth, and tissue differentiation. However, the biological and clinical responses to the same type of hormone treatment differ between subjects with male and female karyotypes. Characterization of the biomolecular mechanisms underlying the sexual dimorphism in response to sex hormones is crucial for the design of new methodological approaches aimed at highlighting sex differences in the patho/physiological aspects caused by the use (and abuse) of sex hormones. To date, in vivo and ex vivo studies are very limited, mainly for ethical reasons; it is therefore of fundamental importance to use experimental approaches based on cellular models at the preclinical level. On the other hand, most in vitro studies do not take into account the chromosomal structure of the cell models used, with the result that ‘sex’ is often a ‘confounding factor’. This highlights the need for basic research to develop preclinical models that take into account chromosomal sex differences to ensure the transferability of the data obtained in vivo and to adequately design relevant experimental studies. Therefore, given the role of skeletal muscle in biological responses to testosterone, the aim of this project is to assess the presence of sex differences in endocrine–metabolic cellular responses to increasing doses of testosterone by simulating functional modifications of skeletal muscle cells in vitro.

**Abstract:**

Muscle tissue is an important target of sex steroids, and particularly, testosterone plays essential roles in muscle cell metabolism. Wide ranges of studies have reported sex differences in basal muscle steroidogenesis, and recently several genes have been identified to be regulated by androgen response elements that show innate sex differences in muscle. However, studies accounting for and demonstrating cell sexual dimorphism in vitro are still scarce and not well characterized. Here, we demonstrated the ability of 46XX and 46XY human primary skeletal muscle cells to differently activate steroidogenesis in vitro, likely related to sex-chromosome onset, and to differently induce hormone release after increasing doses of testosterone exposure. Cells were treated with testosterone at concentrations of 0.5, 2, 5, 10, 32, and 100 nmol/L for 24 h. Variations in 17β-HSD, 5α-R2, CYP-19 expression, DHT, estradiol, and androstenedione release, as well as IL6 and IL8 release, were analyzed, respectively, by RT-PCR, ELISA, and luminex-assay. Following testosterone treatments, and potentially at any concentration level, an increase in the expression of 17β-HSD, 5α-R2, and CYP-19 was observed in 46XY cells, accompanied by elevated levels of DHT, androstenedione, and IL6/IL8 release. Following the same treatment, 46XX cells exhibited an increase in 5α-R2 and CYP-19 expression, a conversion of androgens to estrogens, and a reduction in IL6 and IL8 release. In conclusion, this study demonstrated that sex-chromosome differences may influence in vitro muscle cell steroidogenesis and hormone homeostasis, which are pivotal for skeletal muscle metabolism.

## 1. Introduction

To date, the new challenge for biomedical research is to identify molecular and cellular mechanisms underlying sexual dimorphism [1,2]. Indeed, an ever-growing body of data highlights clear differences in the clinical field, linked to gender and sex, regarding body composition [3], gene expression [4], human physiology [5,6,7], incidence of pathologies, and the response to pharmacological treatment [6,8,9,10,11,12]. Numerous factors can contribute to sexual dimorphism, such as differences in genetics linked to X and Y chromosomes, organ and tissue growth, and differences in sex hormones and sex hormone receptor availability [2,13]. Thus, although sex differences are not limited to the cellular mechanisms and gene expression, different studies observed that only the presence of one more X chromosome, could determine differences in several biological responses, regulating the expression of both autosomal and sex chromosome-linked genes. In humans, 15–25% of X-linked genes are known to escape X-inactivation, emphasizing sexual dimorphism and phenotypic and disease risk variability [14,15,16].

Thus, considering the use of male (XY) and female (XX) primary cells in vitro is pivotal to guarantee the reproducibility of the research and to speculate possible effects in vivo.

Recently, we have demonstrated that male (46XY) and female (46XX) skeletal muscle cells were able to produce sex hormones [17,18] as well as showed differences in the expression of 17β-hydroxysteroid dehydrogenase (17β-HSD), 5α-reductase (5α-R2), aromatase (CYP-19), involved in the steroidogenic process, androgen receptor (AR), and importantly, a different sensitivity of AR following testosterone exposure [19]. However, the effects of differential responsiveness in hormone synthesis of 46XX and 46XY cells exposed to testosterone have not yet been characterized. Thus, investigating in vitro cell processes concerning steroidogenic aspects, activated following the exposure of female and male skeletal muscle cells to physiological/non-physiological testosterone concentrations, represents a pivotal forward step for translational research to improve clinical approaches and pharmacological therapies in gender medicine.

Steroidogenesis (Figure 1) is the process by which cholesterol is converted to steroid hormones. It occurs in steroidogenic tissues according to tissue type and the expression of steroidogenic enzymes [20]. In both males and females, sex steroid hormones are synthesized by 3β-HSD and 17β-HSD, producing progesterone, androstenedione, and testosterone from cholesterol. Testosterone is irreversibly converted to estrogen by aromatase (CYP-19) and into its bioactive metabolite dihydrotestosterone (DHT) by 5α-R2 [21]. These hormones are primarily secreted by the ovary, testis, and adrenal gland, as well as by other peripheral tissues, such as bone, liver, brain, and muscle [22,23]. Particularly, muscle tissue’s ability to synthesize and metabolize sex hormones has been demonstrated [21,24,25,26], and recent observations indicate sex-chromosome differences in skeletal muscle cells’ steroidogenic enzyme gene expression [19].

The present study examines the relationship between hormone balance and the molecular markers associated with muscle metabolism in response to different concentrations of exogenous testosterone in neonatal 46XY and 46XX human primary skeletal muscle cells. In particular, differences related to sexual dimorphism in the variation of 17β-HSD, 5α-R2, CYP-19 expression, DHT, estradiol, and androstenedione release, as well as interleukin (IL)6 and IL8 release, have been investigated. This study aims to provide some molecular evidence for steroid hormone biosynthesis supporting the intrinsic cell ability to respond to specific hormonal environments, pivotal in the activation of different sex-chromosome-specific biochemical processes.

## 2. Materials and Methods

### 2.1. Cell Cultures and Treatments

Prepuberal immortalized human primary skeletal muscle cells purchased from ATCC (PCS-950-010TM, Manassas, VA, USA) and LONZA (LOCC2580, Basel, Switzerland) were cultured in mesenchymal stem cell basal medium (ATCC PCS-500-030) added to a Skeletal Muscle Cell Growth Kit (ATCC PCS-950-040, Lonza LOCC3245). Cells were split 1:2 once weekly and fed 24 h before each experiment. For each experiment, cells were treated for 24 h with increasing doses of testosterone (T) (Sigma Aldrich, St. Louis, MO, USA). After treatment, cells were washed with phosphate-buffered saline (PBS), harvested using trypsinization, and processed according to each protocol procedure. Testosterone (T) was dissolved in methanol at the concentrations of 0, 0.5, 2, 5, 10, 32, and 100 nmol/L, reproducing different adult physiological (female: 0.5–2 nmol/L; male: 10–32 nmol/L) and non-physiological (female: >2 nmol/L; male < 10 nmol/L and >32 nmol/L) serum total testosterone [27,28]. For each experiment, control cells referred to methanol without T. The concentration of methanol is the same for all the different dilutions. For each experiment, three different lots of 46XX and 46XY cells were used. The cells were prepuberal and had never been exposed to a sexual steroid hormone environment before manipulation.

### 2.2. Measurement of Released Hormones

The concentrations of estradiol, DHT, and androstenedione were determined via enzyme-linked immunosorbent assay using commercially available kits, from R&D Systems, DRG International Inc. (Minneapolis, MN, USA), and MyBiosource, Inc. (San Diego, CA, USA), respectively. All samples were processed according to the manufacturer’s protocol and analyzed in duplicate within the same assay.

### 2.3. Cytokines Assay

The supernatant obtained from 2 × 10^4^ treated cells was assayed for IL6 and IL8 using the magnetic bead-based multiplex assay according to the manufacturer’s protocol and as previously described [29,30]. Data acquisition was performed by the Bio-Plex 200 System™ (Bio-Rad Laboratories, Inc., Hercules, CA, USA). Data analysis was performed by Bio-Plex Manager™ 6.0 software. Quality control pools of low, normal, and high concentrations for all parameters were included in each assay. Data were expressed as pg/mL. Cell supernatants were run in triplicate.

### 2.4. RNA Extraction, Reverse Transcription, and Real-Time Quantitative PCR

Total RNA was obtained from ≈3.5 × 10^4^ treated cells using TRIZOL according to the manufacturer’s instructions. cDNA was obtained by reverse transcription of 500 ng of total RNA. RT-qPCRs were performed as previously described [31,32]. Fluorescence intensities were analyzed using the manufacturer’s software (7500 Software v2.05), and relative mRNA abundance was evaluated using the 2^−∆Ct^ method and normalized with *β-ACTIN*. Data were expressed as 2^−∆∆Ct^. Sequences of primers are shown in Table 1.

### 2.5. Statistical Analysis

All data are expressed as means ± SE of three independent experiments, each performed in triplicate. For hormones, myokines, and RNA analysis, an ANOVA test with Bonferroni post hoc analyses was used to determine significant variations (both increasing and decreasing) comparing each T treatment to the control group (ctr) in both 46XX and 46XY cells (marked with an asterisk, *) and comparing 46XX and 46XY to each other at baseline and after each T treatment (marked with octothorpe, #). *p* < 0.05 was accepted as significant. GraphPad PRISM 9 software (GraphPad Software 225, Boston, MA, USA) was used for statistical analysis.

## 3. Results

### 3.1. Sex Dimorphism in Steroidogenic Enzyme Expression and Biosynthesis in Male and Female Human Muscle Cells Exposed to Different Doses of Testosterone

In 46XX and 46XY muscle cells, we analyzed the expression of 5α-R2, CYP-19, and 17β-HSD after increasing doses of T and the amount of DHT, estradiol, and androstenedione in cell medium. As shown in Figure 2A, T treatment induced a high increase in CYP-19 mRNA in 46XY and 46XX cells at all doses analyzed. In particular, in 46XY, cell gene expression was increased by 2.5 ± 0.3-fold with 0.5 nmol/L of T; 135.0 ± 47.1 with T 2 nmol/L; 263.1 ± 1.4 with T 5 nmol/L; 332.6 ± 50.4 with T 10 nmol/L; 128.6 ± 25.7 with T 32 nmol/L; and 49.6 ± 8.3 with T 100 nmol/L. In 46XX, CYP-19 mRNA increased by 11.7 ± 2.3-fold with T 0.5 nmol/L; 301.2 ± 35.8 with T 2 nmol/L; 808.3 ± 30.1 with T 5 nmol/L; 801.8 ± 94.2 with T 10 nmol/L; 320.4 ± 34.7 with T 32 nmol/L; and 80.0 ± 6.6 with T 100 nmol/L (** *p* < 0.01, all treatment concentrations vs. untreated cells). Furthermore, in 46XX cells, CYP-19 mRNA increased more strongly than in 46XY cells at almost each dose analyzed (## *p* < 0.01 46XX vs. 46XY cells). Concerning 5α-R2 (Figure 2B), we observed a significant decrease in 5α-R2 mRNA in 46XX cells of 0.7 ± 0.2 with T 2 nmol/L (** *p* < 0.01, vs. untreated cells) and a significant increase of 1.6 ± 0.0 with T 5 nmol/L; 1.3 ± 0.1 with T 10 nmol/L; 1.4 ± 0.1 with T 32 nmol/L and 1.6 ± 0.1 with T 100 nmol/L (** *p* < 0.01, vs. untreated cells); conversely, in 46XY, we observed an increase in 5α-R2 mRNA only after T 0.5 nmol/L (1.4 ± 0.0 vs. ctr) (***p* < 0.01, vs. untreated cells), and a significant decrease of 0.7 ± 0.0-fold with T 2 nM; 0.8 ± 0.0-fold with T 5 nmol/L; 0.6 ± 0.1-fold with T 10 nM; 0.4 ± 0.1-fold with T 32 nmol/L; and 0.2 ± 0.1-fold with T 100 nmol/L (** *p* < 0.01, vs. untreated cells) (Figure 2B).

As shown in Figure 2C (white columns), in 46XX cells, T induced a rising increase in estradiol concentration in comparison to untreated cells, by 1.2 ± 0.1-fold with T 0.5 nmol/L; by 1.9 ± 0.1-fold with T 2 nmol/L; 1.8 ± 0.2-fold with T 5 nmol/L; 2.7 ± 0.2-fold with T 10 nmol/L; 2.3 ± 0.1-fold with T 32 nmol/L; and 2.9 ± 0.1-fold with T 100 nmol/L (** *p* < 0.01 vs. untreated cell). Differently, in 46XY cells (Figure 2C black columns), we observed no significant effects of T at 0.5, 2, and 5 nmol/L, and a slight but significant decrease in estradiol release after T 10 (0.8 ± 0.0 vs. ctr, * *p* < 0.05), 32 (0.6 ± 0.0 vs. ctr, * *p* < 0.05), and 100 nmol/L (0.5 ± 0.0 vs. ctr, * *p* < 0.05). Concerning DHT secretion (Figure 2D), in comparison to untreated cells, we observed a significant increase in hormone release in both 46XX (1.3 ± 0.0-fold with T 0.5 nmol/L; 1.4 ± 0.0-fold with T 2 nmol/L; 1.7 ± 0.2-fold with T 5 nmol/L; 5.2 ± 0.1-fold with T 10 nmol/L; 6.5 ± 0.1-fold with T 32 nmol/L; and 25.0 ± 0.3-fold with T 100 nmol/L) and 46XY cells (1.2 ± 0.0-fold with T 0.5 nmol/L; 2.1 ± 0.1-fold with T 2 nmol/L; 3.6 ± 0.3-fold with T 5 nmol/L; 3.7 ± 0.5-fold with T 10 nmol/L; 6.0 ± 0.1-fold with T 32 nmol/L; and 17.2 ± 0.5-fold with T 100 nmol/L). Interestingly, we observed that, although at the lowest doses of 2 and 5 T nmol/L, in 46XY cells DHT release was significantly higher in comparison to 46XX, at the higher doses of 10 and 32 nmol/L, 46XX cells reached the same levels of DHT achieved by male cells and exceeded DHT levels of 46XY cells when exposed to T 100 nmol/L (Figure 2D).

### 3.2. Increase in 17β-Hydroxysteroid Dehydrogenase Expression and Androstenedione Synthesis in 46XY Cells Following Testosterone Exposure

As previously observed, in our experimental conditions, 46XX cells showed an undetectable amount of 17β-HSD mRNA at the basal level that, differently, was expressed in 46XY muscle cells [19]. For this reason, we analyzed the variation of 17β-HSD mRNA after T exposure only in 46XY cells. We observed that T treatment increased significantly 17β-HSD mRNA by 3.2 ± 0.9 fold after 0.5 nmol/L, by 3.6 ± 0.5 fold after 2 nmol/L, and by 2.2 ± 0.3 fold after 5 nmol/L; conversely, at 10 nmol/L, 32 nmol/L, and 100 nmol/L of T, we observed a significant decrease of 0.6 ± 0.1 fold, 0.5 ± 0.1 fold, and 0.7 ± 0.0 fold, respectively, in comparison to the untreated condition (Figure 3A).

Finally, we observed a proportional increment of androstenedione release in 46XY cells with increasing T exposure (Figure 3B), by 1.2 ± 0.0-fold with T 5 nmol/L; 1.5 ± 0.2-fold with T 10 nmol/L; 2.2 ± 0.3-fold with T 32 nmol/L; and 2.7 ± 0.7-fold with T 100 nM.

### 3.3. Sex Dimorphism in IL6 and IL8 Release in Male and Female Human Muscle Cells Exposed to Different Doses of Testosterone

Finally, we analyzed the release of IL6 and IL8 myokines. As shown in Figure 4A,B, 46XX cells, at basal condition, showed a greater amount of both myokines in comparison to 46XY cells (respectively, for IL6, 396.4 ± 14.8 vs. 4.3 ± 0.3, ## *p* < 0.01; for IL8, 6027.2 ± 436.2 vs. 251.2 ± 13.9; ## *p* < 0.01). However, in 46XY cells, T induced a significant increase in IL6 secretion after 0.5, 2, 5, 10, 32, and 100 nmol/L of treatment (Figure 4A, black columns, and Appendix A), and of IL8 after 10, 32, and 100 nmol/L of treatment (Figure 4B, black columns, and Appendix A). Conversely, in 46XX cells, T tended to increase both IL6 and IL8 release at 0.5 nmol/L of treatment, whereas decreased myokines release after 2, 5, 10, 32, and 100 nmol/L of treatment (Figure 4A,B, white columns, and Appendix A).

## 4. Discussion

In this study, we analyzed sexual dimorphism in muscle steroidogenesis in 46XY and 46XX human skeletal muscle cells exposed to increasing doses of testosterone. We observed a different cell responsiveness concerning hormone release and steroidogenic enzyme expression. We showed that sex dimorphism in skeletal muscle occurs in vitro and is probably related to X or Y chromosome presence. Specifically, we found that 46XY cells were inclined to produce DHT and androstenedione, likely to minimize estrogen conversion and maximize intramuscular androgen levels. In contrast, 46XX muscle cells probably converted androgens to estrogens, reducing intramuscular androgen levels. Furthermore, we observed that this difference in hormone release may be related to the distinct role of androgen and estrogen in regulating muscle metabolism and skeletal–muscle tissue homeostasis.

The synthesis of the sex steroid hormones progesterone, androstenedione, and testosterone from cholesterol represents a key step in the process of steroidogenesis. The process occurs in both men and women, primarily in the gonads and adrenal glands, as well as in other peripheral tissues, including bone, liver, brain, and muscle [22,23]. Moreover, recent findings have revealed differences between sex chromosomes in the gene expression of steroidogenic enzymes in skeletal muscle cells [19].

In this study, we observed that in female cells, exposure to T at the lowest doses (0.5–2 nmol/L, corresponding to a physiological condition in females) notably increased both aromatase expression and estradiol secretion (Figure 2A,C). Conversely, at higher doses mimicking female hyperandrogenism, testosterone favored conversion DHT, reaching levels comparable to or exceeding those observed in males at the highest T exposure (Figure 2D). Additionally, the mRNA expression of 5α-R2, crucial for converting testosterone to DHT, was up-regulated (Figure 2B). Thus, under physiological androgen conditions in 46XX cells, muscle anabolism and catabolism processes involve estrogen activity. These findings align with previous literature demonstrating that estrogens in females, like androgens in males, contribute to anabolic and catabolic effects on muscle [33,34,35,36]. Furthermore, besides the role in controlling muscle metabolism, estrogens in muscle cells exert other actions via paracrine/endocrine mechanisms, regulating myokine release, bone homeostasis, vascular tone, and smooth muscle cell metabolism [37,38,39,40]. In this context, we observed that 46XX cells significantly decreased the release of IL6 and IL8 myokines (Figure 4A,B).

Conversely, despite their role as pro-inflammatory cytokines [29,41], IL6 and IL8 exert important effects on muscle cell metabolism, acting as “energy sensors” and modulating carbohydrate availability, glucose oxidation, and lipolysis, particularly during muscle contraction [42]. These myokines may be released in large amounts from working muscles into the bloodstream, where they can exert pleiotropic hormone-like effects in other organs. Previous evidence indicates that IL6 and IL8 play a role in bone resorption, stimulating the increase in CFU-granulocyte macrophages (CFU-GM) and inducing bone resorption [43,44]. Therefore, the increase in estradiol may act as an inhibitor of the induction of IL6 and IL8 by muscle cells to guarantee bone protective effects [43,44]. On the contrary, we observed that 46XY cells significantly increased both IL6 and IL8 release. Probably, in males, T primarily engages in signal transduction processes involved in cell metabolism, strength, and energy expenditure. Furthermore, androgens increase Ca^2+^ mobilization and muscle contraction [45], mirroring tissue contraction during physical exercise, responsible for IL6 and IL8 releases [42].

As previously mentioned, when stimulated with higher doses of T, in 46XX cells, we observed high release of DHT. Thus, we can suggest that in female hyperandrogenism conditions (T > 2 ng/mL), the higher levels of circulating T may effortlessly activate androgen receptors in female muscle and direct the steroidogenic process towards androgen synthesis. This activation is easily induced in 46XX cells, where we have already demonstrated there is a greater sensitivity of AR compared to male cells to respond at lower levels of T [19]. Furthermore, these data confirm the clinical observations in the literature, indicating the increase in free testosterone and DHT in hyperandrogenic female patients affected by Polycystic Ovary Syndrome (PCOS) [46,47,48] and highlighting the already discussed themes concerning the possible performance advantage of transgender women in the female sport category [49,50,51,52].

Unlike females, 46XY cells maintained an androgen environment subsequent to each dose of androgen stimulation, promoting DHT and androstenedione synthesis. In fact, from the lowest doses of T, we observed an increase in DHT secretion (Figure 2D) but no increment of estradiol secretion (Figure 2C). However, also in male cells, we observed an important upregulation in the expression of CYP-19 mRNA (Figure 2A) and a significant decrease in 5α-R2, inversely related to T concentration (Figure 2B). 5α-R2 is largely expressed in males [19] and probably right translated in its protein form. The observed DHT secretion probably translated correctly into its protein form. The observed DHT secretion probably relates to an enzyme amount already present in 46XY cells and then restored, with mRNA translation, with increasing doses of testosterone. Concerning CYP-19, we suggest that 46XY cells convert T to androstenedione (Figure 2A,B) and then possibly to estrone by the aromatase enzyme. Estrone is a weak estrogen agonist, and although, like other types of estrogen, it plays a role in female sexual function, it is not as powerful as other types of estrogen [53]. It is mainly produced after menopause by the adrenal glands and fatty tissue and behaves as a storehouse for estrogen, being converted into estrogen when needed. Estrone shows a low affinity for the estrogen receptor compared with estradiol [53], but earlier studies showed that when present in excess (ratio estrone/estradiol more than 4-fold) it can interfere with the ability of estradiol to bind its receptor, thus acting as an estrogen antagonist [54].

Previously, in vivo and in ex vivo studies have discussed sexual dimorphism in muscle cells and the diverse roles of estrogen and androgen receptors in muscle metabolism well [55,56]; here, we report a possibly different response of 46XY and 46XX human skeletal muscle cells to the same hormone treatment, activating sex-specific hormone biosynthesis (Figure 5).

## 5. Conclusions

In conclusion, in this observational study, we examined sex-dimorphism in muscle steroidogenesis after testosterone exposure. In particular, at physiological concentrations, circulating testosterone may preferentially induce the formation of DHT and androstenedione in 46XY skeletal muscle cells, whereas it may induce the formation of estradiol in 46XX cells. In contrast, under hyperandrogenic conditions, testosterone is likely to induce the release of DHT in 46XY cells and the release of estradiol and DHT in 46XX cells (Figure 5). Although some new and interesting observations have been performed, this study shows some limitations. First, the number of cell donors. Even though the experiments were performed at least three times in triplicate with three different lots of cells, they were derived from two male and female donors due to the difficulty in obtaining neonatal primary cells. Thus, three replicates with another donor may be advisable to carry on further investigations. Additionally, we did not show evidence if the sex dimorphism in hormone metabolism is still observed in myotubes. In this context, further experiments in differentiated conditions analyzing the molecular mechanisms responsible for this dimorphism are now in progress. In this context, the use of antagonists, agonists, and/or receptor inhibitors will be necessary to affirm which pathway is really favored over another. Among various reasons, it is known that hormone response in muscle is regulated by hundreds of miRNAs and non-coding RNAs that are mainly situated on the X chromosome [57]. Although it is true that in females one of the X chromosomes is inactivated, it is also demonstrated that part of the DNA can escape this mechanism [14,15,16,58,59]. Moreover, hormones exert strong influences on muscle, controlling proliferation, growth, and survival. Further studies investigating both the role of non-coding RNA in muscle endocrinology and/or other released molecules involved in muscle metabolism, could be a useful starting point to completely understand the dimorphism in cell response related to sex chromosomes.

## Figures and Tables

**Figure 1 biology-13-00796-f001:**
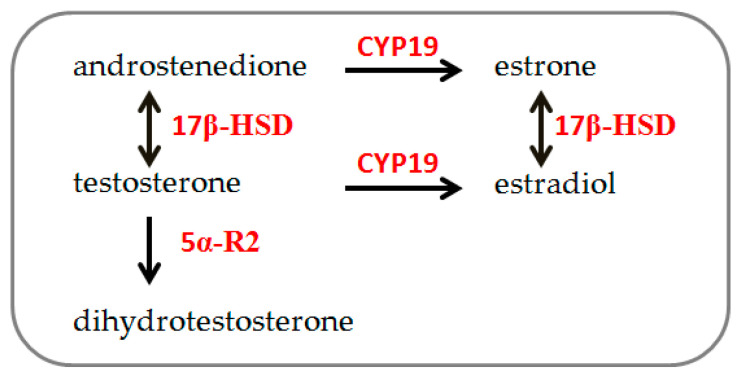
Schematic simplified description of steroidogenesis pathway: 17β-HSD: 17β-hydroxysteroid dehydrogenase; 5α-R2: 5α-reductase. CYP-19: aromatase (modified from [19]).

**Figure 2 biology-13-00796-f002:**
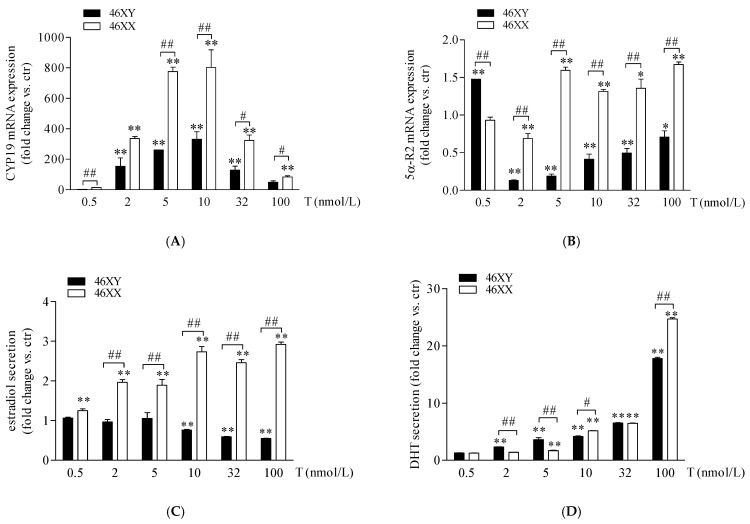
CYP-19 (**A**) and 5α-R2 (**B**) mRNA expression in 46XY (black columns) and 46XX (white columns) human primary human muscle cells. Data are shown as fold induction (2^−ΔΔct^) ± SE (n = 3). Estradiol (**C**) and DHT (**D**) release in 46XY (black columns) and 46XX (white columns) human primary human muscle cells. Data are shown as fold increase vs. ctr taken as 1 ± SE (n = 3). * *p* < 0.05 and ** *p* < 0.01 46XY and 46XX cells vs. ctr; # *p* < 0.05 and ## *p* < 0.01 of 46XY vs. 46XX cells. Statistical significance was determined with ANOVA with Bonferroni’s post hoc test. CYP-19 = aromatase; 5α-R2 = 5α-reductase.

**Figure 3 biology-13-00796-f003:**
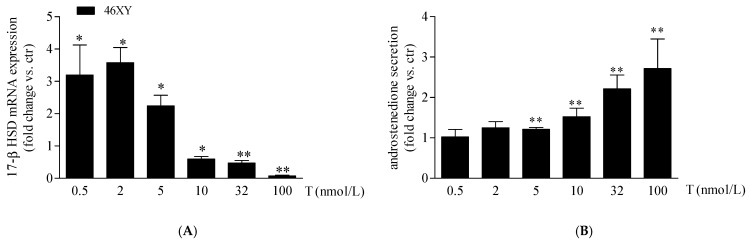
17β-HSD mRNA (**A**) and androstenedione release (**B**) in 46XY human primary human muscle cells. Data are shown as fold increase vs. ctr taken as 1 ± SE (n = 3). * *p* < 0.05 and ** *p* < 0.01 46XY cells vs. ctr. Statistical significance was determined with ANOVA with Bonferroni’s post hoc test. 17β-HSD = 17β-hydroxysteroid dehydrogenase.

**Figure 4 biology-13-00796-f004:**
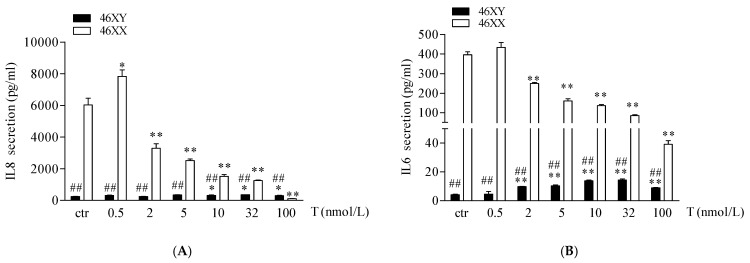
IL6 (**A**) and IL8 (**B**) release in 46XY (black columns) and 46XX (white columns) human primary human muscle cells (exposed to different concentrations of testosterone). Data are presented as the means ± SE (n = 3). * *p* < 0.05 and ** *p* < 0.01 vs. relative control within group; ## *p* < 0.01 vs. corresponding treatment between groups. Statistical significance was determined by an ANOVA with Bonferroni’s post hoc test.

**Figure 5 biology-13-00796-f005:**
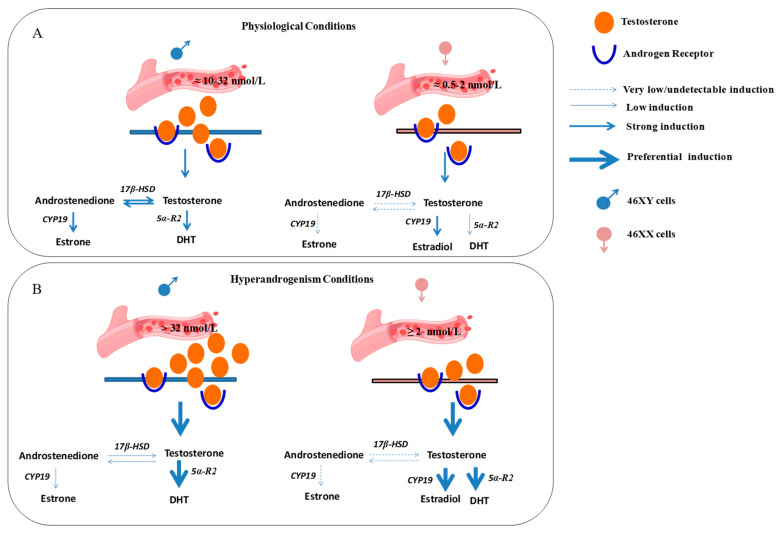
Possible differential hormone synthesis in 46XX and 46XY muscle cells after testosterone treatment. Androgens from blood vessels may act in muscle cells via trans-membrane and/or cytoplasmic androgen receptors and activate differently steroidogenic processes according to sex-chromosome differences. (**A**) Circulating androgens at physiological concentration activate hormone synthesis in skeletal muscle cells preferentially toward DHT and androstenedione formation in 46XY and estradiol formation in 46XX cells. (**B**) In hyperandrogenic conditions, 46XY cells strongly activate steroidogenesis to release DHT, whereas 46XX cells stimulate high estradiol and DHT synthesis. 17β-HSD: 17β-hydroxysteroid dehydrogenase; 5α-R2: 5α-reductase; CYP-19: aromatase; 5α-R2: 5α-reductase; DHT: dihydrotestosterone.

**Table 1 biology-13-00796-t001:** Sequences of primers for RT-PCR analysis.

Gene Name	Acronym	Forward 5′–3′	Reverse 5′–3′
5α-reductase	*5α-R2*	AGTGGAGGGCATGGTGCTAA	TCTCTCACTTAGCACGGGGA
17β-hydroxysteroid dehydrogenase	*17β-HSD*	TTTGCGCTCGAAGGTTTGTG	GCAGTCAAGAAGAGCTCCGT
Aromatase	*CYP-19*	ATGTTTCTGGAAATGCTGAAC	CTGTTTCAGATATTTTTCGCTG
β-ACTIN	*ACTB*	AAC CTGAACCCCAAGGCC	AGCCTGGATAGCAACGTACA

## Data Availability

Not applicable.

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
