# Peer review of "Sexual Dimorphism in Sex Hormone Metabolism in Human Skeletal Muscle Cells in Response to Different Testosterone Exposure"

_biology, 2024, doi:10.3390/biology13100796_

Round 1

Reviewer 1 Report

Comments and Suggestions for Authors

Comments to the Author

The communication, “Sex dimorphism in hormone expression and release in human 2 skeletal muscle cells in response to different testosterone exposure”, provides valuable insights into sexual dimorphism in muscle steroidogenesis. Specifically, this study revealed major differences in enzymes involved in steroidogenesis and hormone release to testosterone exposure in 46XY and 46XX human primary skeletal muscle cells. This research outlines a novel approach to new methodologies for determining sex differences in response to hormone supplementation. This is very important and novel work that significantly contributes to the field.

Title: “Sex dimorphism” should be changed to “Sexual Dimorphism” “hormone expression” doesn’t make sense to me. You did not measure expression of hormones, you measured hormone release and expression of enzymes involved in steroidogenesis. Your title should be more specific to the study (i.e. what about the myokines analyzed or enzymes involved in steroidogenesis?)

Simple Summary:

Line Comment

21 “Characterize” should be “Characterization of”

22 “design of new”

45 Avoid use of first person “We” throughout manuscript

46 After all treatment concentrations? If so, this should be specified.

47 The sentence, “Conversely, 46XX cells increased the expression of 5α-R2 and CYP-19, preferentially converted androgens to estrogens, and kept unchanged the release of IL6 and IL8.” The use of “Conversely” makes it sound like the results differ from that of the 46XY cells, when this is not the case for expression of 5α-R2 and CYP-19, as both are increased. Additionally, the sentence is worded to sound like the cells caused the increased expression, when it was the treatments of testosterone. Lastly, the tail end of the sentence that states, “and kept unchanged the release of” does not make sense. This part of the sentence also needs restructuring. Perhaps, “ and release of IL6 and IL8 were unchanged.” could work?

49 “sex-chromosomes differences” - This should be “sex-chromosome differences”?

50 reword to “hormone homeostasis, which are pivotal”

Abstract:

Line Comment

Keywords: Typically, these are organized alphabetically.

Introduction:

Line Comment

58 “in the clinical field”

63 “receptor”

64 “gene expression”

64 “only the presence”

65 “X chromosomes”

66 “sex chromosome-linked”

68 “sexual dimorphism and phenotypic and disease risk”

75 “Figure 1” - capitalize the first letter of “Figure” and “Table” throughout manuscript

80 “non-physiological”

82 “clinical approaches”

82 “therapies”

Materials and Methods:

96 “Cell cultures” - need consistency with capitalization of these titles throughout manuscript

97 What is eugenetic? - I’m unfamiliar with this term and am not seeing anything in the literature. More information is needed on how cells were harvested/obtained.

100 Should not start a sentence with an abbreviation - follow this throughout the manuscript

103 “For each experiment, control cells”

103 Was the concentration of methanol the same as T in the T treated cells? If so, make this clear.

104 “For each experiment, three”

106 “steroid hormone environment”

110 Need city and state of R&D systems in parentheses after it’s listed

111 Should be “(San Diego, California, USA)”

109 move “, respectively” to the end of the sentence

114 “supernatant obtained from 2 x”

116 references are out of order. Need to fix this throughout the manuscript

120 “pg/mL”

120 “Cell supernatants”

126 “mRNA abundance was evaluated”

131 Data should be expressed as means ± standard error. Why weren’t standard errors used? 

134 It is unclear whether 46XX and 46XY are compared to each other AND compared to their baseline measurements? Statistics section needs to be re-written to provide clarity to the readers.

135 Space needed before “The GraphPad PRISM” sentence starts.

NOTE: I stopped editing for English language grammar after the Methods section. This manuscript will need significant English grammatical editing before it can be considered for publication. Many spelling mistakes and grammatical errors throughout.

Results:

149 it’s unclear what this p-value is for? Does this mean all treatment concentrations were significantly increased when compared to their respective untreated counterpart? P-values needed after mentions of “increase” and “decrease” to specify what the p-values pertain to? Clarity is needed in both the Methods and Results section.

215 “implicated in energy metabolism, muscle growth and proliferation” should be removed from the results section. The results section should only state results. This should be moved to the discussion section.

Discussion:

247 This paragraph would be better suited in the introduction to give the reader a brief background of WHY these enzymes were chosen and their role in steroidogenesis. This information is good to include in the discussion too, but to a lesser extent.

309 So is this sentence saying that estrone is an agonist or antagonist of estrogen? The two references (49 and 50) seem to contradict each other? This sentence may need reworded to provide clarity.

Overall: The discussion is very well structured and links the research findings of the present study to previously observed results from past studies. The author’s do a great job with describing what their results may mean and how their research can be applied.

Conclusions:

Overall: I think you should sum up your results and conclusions from those results as a last sentence to bring everything full circle.

Tables:

128 Table 1: needs to include the full gene names as a footnote in the table, as each table should be able to stand alone.

128 Table 1: for the CYP-19 line, we need more space between the forward and reverse primers because I can’t tell where they end/start.

Figures:

185 Figure 2: text font looks different on y axes of some. 

185 Figure 2: Definitions for abbreviations of analyzed enzymes needs to be included in the figure legend, as each figure should be able to stand on its own. Same comments for Figures 3, 4, and 5.

190 Figure 2: Meaning of red dotted line is unclear? Unsure what “control with control vehicle take as (ctr) 1 means”? Same comments for Figure 3.

References:

Overall: These seem to be out of order in the text. Please double check this.

Comments on the Quality of English Language

NOTE: I stopped editing for English language grammar after the Methods section. This manuscript will need significant English grammatical editing before it can be considered for publication. Many spelling mistakes and grammatical errors throughout.

Reviewer 2 Report

Comments and Suggestions for Authors

The manuscript explores the differential responses of muscle cells from males (46XY) and females (46XX) to testosterone. The authors treated these cells with various concentrations of testosterone and measured changes in specific gene expressions and hormone levels. They reported differences in gene expression and hormone release between the sexes.

Major Issue

The manuscript has a significant limitation: it utilizes only one male and one female biological replicate. Without at least three biological replicates, it is challenging to determine whether the reported differences are genuinely attributable to sex rather than variations between individual cell lines. The authors should include more biological replicates to strengthen their findings. Simply, discussing this major limitation is not sufficient. 

Minor Issues

Terminology: The authors should use "X chromosome" instead of "chromosome X" for clarity and consistency throughout the text.

Line 18: Replace "(i.e., testosterone)" with "(e.g., testosterone)" for accuracy.

Line 67: The phrase "In fact, chromosome X is enriched..." should be replaced with "In humans, 15-25% of X-linked genes are known to escape X-inactivation." The term "enriched" is not accurate in this context.

Comments on the Quality of English Language

 Overall, the manuscript is well-written.

Round 2

Reviewer 1 Report

Comments and Suggestions for Authors

Comments to the Author

The authors did well incorporating feedback into a much stronger version of the original manuscript. There is further moderate level English editing required (example: comma use throughout, consistency of capitalization of titles, p-values, and the like).

Introduction:

Line Comment

64 Noticing that the sentence still doesn’t quite flow. Recommend removing “the” before “only” and removing the “s” at the end of “chromosomes”. Recommend rewording to “different studies observed that only the presence of one more X chromosome, could”

Materials and Methods:

108 Still need consistency with capitalization of these titles all throughout manuscript

109 Eugenetic appears to still be spelled incorrectly.

120 remove “s” at end of “experiment”

122 remove “s” at the end of “experiment”

129 Still should be “(San Diego, California, USA)”

132 remove “at” from before the concentration of cells stated

Results:

162 capitalize “figure”

162 change “an” to “a”

163 move comma to after “cells”

169 consistency with p-value spacing needed throughout the manuscript (for example: p<0.01 vs p < 0.01). Pick one and be consistent throughout. 

171 was 5α-R2 increase significant? I’m assuming so, but need to say significant since p-value is not listed and we describe a significant change as P < 0.05.

178 capitalize “figure”

183 was this decrease in estradiol significant?

210 remove period from end of 3.2 title to be consistent

218 “respectively” is misplaced. It should be at the end of the fold changes

223 period missing at end of sentence

Tables:

146 Table 1: acronym for beta-actin is missing

Figures:

248 Figure 4: need consistency with capitalization of p-values in figure legend. 

337 Figure 5: All abbreviated hormones and enzymes need defined in figure legend. Also, capitalize “conditions” in the title of Figure 5b

References:

Overall: These seem to be out of order in the text. Please double check this.

Comments on the Quality of English Language

There is further moderate level English editing required (example: comma use throughout, consistency of capitalization of titles, p-values, and the like).

Reviewer 2 Report

Comments and Suggestions for Authors

The authors added 1 more biological replicate and discussed the limitation of the study suggesting future studies using more replicates. The manuscript is well written and fits the Biology Journal. 
